# Sequence analysis of tyrosine recombinases allows annotation of mobile genetic elements in prokaryotic genomes

Georgy Smyshlyaev[1,2,3,*] (iD), Alex Bateman[1] (iD) & Orsolya Barabas[2,3] (iD)

## Abstract

**Mobile genetic elements (MGEs) sequester and mobilize antibiotic resistance genes across bacterial genomes. Efficient and reliable identification of such elements is necessary to follow resistance spreading. However, automated tools for MGE identification are missing. Tyrosine recombinase (YR) proteins drive MGE mobilization and could provide markers for MGE detection, but they constitute a diverse family also involved in housekeeping functions. Here, we conducted a comprehensive survey of YRs from bacterial, archaeal, and phage genomes and developed a sequence-based classification system that dissects the characteristics of MGE-borne YRs. We revealed that MGE-related YRs evolved from non-mobile YRs by acquisition of a regulatory arm-binding domain that is essential for their mobility function. Based on these results, we further identified numerous unknown MGEs. This work provides a resource for comparative analysis and functional annotation of YRs and aids the development of computational tools for MGE annotation. Additionally, we reveal how YRs adapted to drive gene transfer across species and provide a tool to better characterize antibiotic resistance dissemination.**

**Keywords** antibiotic resistance; evolution; horizontal gene transfer; mobile genetic elements; tyrosine recombinases

**Subject Categories** Chromatin, Transcription & Genomics; Evolution & Ecology; Microbiology, Virology & Host Pathogen Interaction

**Mol Syst Biol. (2021) 17: e9880**

## Introduction

Tyrosine recombinases (YRs) form a large family of proteins that perform site-specific DNA recombination in a wide variety of biological processes (Grindley *et al*, 2006; Jayaram *et al*, 2015). They promote post-replicative segregation of plasmids and circular chromosomes upon cell division, thereby protecting genome integrity in bacteria, archaea, and phages. For example, the highly conserved Xer proteins (e.g., XerC and XerD in *E. coli*) resolve chromosome multimers formed after DNA replication in prokaryotes (reviewed in (Midonet & Barre, 2014)), and the Cre recombinase separates dimers of the P1 phage genome (reviewed in Van Duyne, 2015). Other YRs act as genetic switches, triggering phenotype variation within bacterial populations via DNA inversion or deletion (Klemm, 1986; Manso *et al*, 2014; Li *et al*, 2016).

In addition, YRs can drive the movement of mobile genetic elements (MGEs), including phages and transposons. Some MGEs hijack host-encoded Xer proteins (Huber & Waldor, 2002; Das *et al*, 2013), while others encode specific YRs to promote their own integration and transfer in bacterial genomes (Van Houdt *et al*, 2012; Wang *et al*, 2018). Prominent examples of YR-carrying MGEs include the integrative and conjugative elements (ICEs), also referred to as conjugative transposons. These elements combine features of phages and plasmids, because they actively integrate into genomes and disseminate by conjugative transfer (Burrus & Waldor, 2004). ICEs and related mobilizable elements are abundant in bacterial genomes and provide efficient vehicles for horizontal transfer of genetic traits such as virulence and antibiotic resistance (Franke & Clewell, 1981; Harbottle *et al*, 2006; Stokes & Gillings, 2011; Bellanger *et al*, 2014; Guédon *et al*, 2017). Furthermore, YRs are present in antibiotic resistance-carrying non-conjugative transposons (Vanhooff *et al*, 2006; Siguier *et al*, 2014), and they mediate the acquisition and activation of antibiotic resistance genes in complex functional units in integrons (reviewed in (Mazel, 2006)).

Therefore, YRs contribute to the emergence and spread of bacterial virulence and antibiotic resistance in several ways and characterizing their distribution and functions is instrumental to understanding horizontal gene transfer and adaptation in bacteria. Yet, functional annotation of these proteins in genome sequence data has been hindered by their large diversity, leaving the vast majority of YRs without a reliable functional prediction and a comprehensive classification. This limits our understanding of the diversity and phylogenetic relationships of different YR groups and hampers automated identification and functional annotation of YRs and YR-carrying MGE.

---

1   European Molecular Biology Laboratory, European Bioinformatics Institute (EMBL-EBI), Hinxton, UK
2   European Molecular Biology Laboratory (EMBL), Structural and Computational Biology Unit, Heidelberg, Germany
3   Department of Molecular Biology, University of Geneva, Geneva, Switzerland
    *Corresponding author. Tel: +41 22 379 61 84; E-mail: georgy.smyshlyaev@unige.ch

Here, we assembled an extended set of prokaryotic YR sequences. By sequence analysis and classification, we showed that YRs of ICEs and phages emerged from a Xer-like ancestor by acquisition of an additional DNA-binding domain. In turn, short "simple" YRs function in chromosome dimer resolution, integron recombination, or mobilization of non-conjugative elements. YRs within phylogenetic groups share specific characteristics, such as distinctive structural features and preferred host taxonomy. With this YR classification system, we further developed a tool to identify MGEs in genome sequence data and used it to detect and characterize numerous novel elements in diverse bacteria. These results illuminate YR diversity, distribution, and evolution, provide a resource for mapping YR function, and help to characterize the role of these proteins in shaping bacterial genomes and the spread of antibiotic resistance.

## Results

### Identification and analysis of tyrosine recombinases

Previous structural and sequence analyses indicated that YRs generally have two main functional domains: The core-binding (CB) domain binds the recombination DNA site, and the catalytic (CAT) domain catalyzes all DNA cleavage and joining reactions required for recombination (Esposito & Scocca, 1997; Nunes-Düby *et al,* 1998; Swalla *et al,* 2003). Some YRs have an additional N-terminal arm-binding (AB) domain that recognizes accessory DNA sequences, so-called arm sites, near the recombination sites. Crystal structures showed that the CAT domain has a similar fold in diverse YRs (Guo *et al,* 1997; Subramanya *et al,* 1997; Tirumalai *et al,* 1997; Skaar *et al,* 2015) and comparative sequence analyses revealed two highly conserved regions (referred to as boxes) and three patches with less significant conservation (Esposito & Scocca, 1997; Nunes-Düby *et al,* 1998). Conserved regions include the catalytic residues, i.e., the tyrosine nucleophile and the catalytic pentad RKHRH (Jayaram *et al,* 2015), as well as the hydrophobic protein core. The CB domain is much less conserved on the sequence level, but its structural architecture is also preserved (Swalla *et al,* 2003). In turn, the AB domain is highly variable with substantial structural and sequence diversity between YR family members (Clubb *et al,* 1999; Fadeev *et al,* 2009; Szwagierczak *et al,* 2009).

To analyze the diversity of YRs, we employed the following strategy. First, we performed an iterative jackhmmer search against the UniProt reference proteomes database using the prototypical XerD protein from *Escherichia coli* as an initial query. In every cycle of this search, the hit sequences were aligned and a profile hidden Markov model (profile HMM) was built. Profile HMM is a probabilistic model used to describe characteristic sequence features of the alignment. This profile HMM was then used as a new query in the next search cycle. This iterative procedure allows identification of distantly related homologues of the original query (Johnson *et al,* 2010; Potter *et al,* 2018). The resulting sequences were then clustered, and the representatives of the clusters were aligned. The alignment was truncated to contain only the CB and CAT regions, which are ubiquitously present in all YR proteins. This resulting alignment was then used to reconstruct the phylogenetic tree with the PhyML package (Fig 1A and Appendix Fig S1). The tree topology was supported by parametric aBayes and non-parametric SH-LRT tests (Anisimova *et al,* 2011). Based on phylogeny, we then divided YRs into subgroups with significant branch supports (over 0.98 and 0.85 for aBayes and SH-LRT, respectively; Appendix Table S1). For each subgroup, we created a distinctive profile HMM, which we then used to find all YR homologues in the UniProt reference proteomes collection. For the resulting sequences, we created sequence logos to visualize conserved regions within subgroups (Appendix Figs S2–S4) and analyzed the specific differences between subgroups (Fig 2). We mapped all YR proteins to their genomes of origin and tracked the taxonomic distribution of each subgroup (Fig 1B, Dataset EV1). Finally, we extracted the fifty most abundant YR proteins and characterized their distribution, classification, and putative function (Fig 1C, Dataset EV2).

This analysis showed that all YRs can be classified into two major phylogenetic groups: simple YRs, which consist of a CB and a CAT domain, and complex YRs, which contain an additional AB domain (Figs 1A and 2). Within these main groups, smaller subgroups were identified, which share a generally conserved domain architecture, but vary in specific structural and sequence features (Appendix Fig S1). Notably, YRs within subgroups have a characteristic taxonomic distribution and share similar predicted functions. In the following sections, we summarize the key sequence features and functional characteristics of all major groups and subgroups.

### Simple YRs

The first major YR group revealed in our study includes simple YRs. Members of this group usually comprise only CB and CAT domains and can be further classified into fourteen subgroups (Figs 1A and 2, Appendix Fig S1).

The largest subgroup, Xer, mainly contains recombinases that are responsible for chromosome dimer resolution in bacteria and archaea, such as XerC/D, XerH, XerS, and XerA (Carnoy & Roten, 2009; Cortez *et al,* 2010; Nolivos *et al,* 2010; Debowski *et al,* 2012). Sequence comparisons revealed that proteins in this subgroup are highly conserved, with numerous residues conserved also outside of the active site pocket and the hydrophobic core (Appendix Figs S2–S4). The subgroup is widely distributed, and its members are present in almost all analyzed bacterial and archaeal classes (Fig 1B, Dataset EV1), which is consistent with the essential role of these proteins. In the remaining taxa, other class-specific simple YRs may compensate for Xer function. For example, in Halobacteria we found a specific type of simple YRs, named Arch1, which resemble Xer but contain short distinct sequence insertion (Fig 2 and Appendix Fig S3). Similarly, Oscillatoriophycideae lack a Xer protein and instead contain members of the separate Cyan subgroup (named after Cyanobacteria, a phylum of the class). Furthermore, the Cand subgroup unites Xer-related YRs from unclassified "Candidate" phyla, a "microbial dark matter" (Rinke *et al,* 2013).

The remaining subgroups of simple YRs do not seem to mediate essential cellular functions, but are linked to MGEs. Members of the RitA, RitB, and RitC subgroups are found on "recombinases in trio" transposable elements (Ricker *et al,* 2013) and contain added structural features compared with the Xer proteins. RitA YRs have two sequential CB domains, and RitC YRs contain an additional conserved C-terminal helix, both of which are absent in all other YRs (Fig 2 and Appendix Fig S5). The TnpA subgroup unites YRs

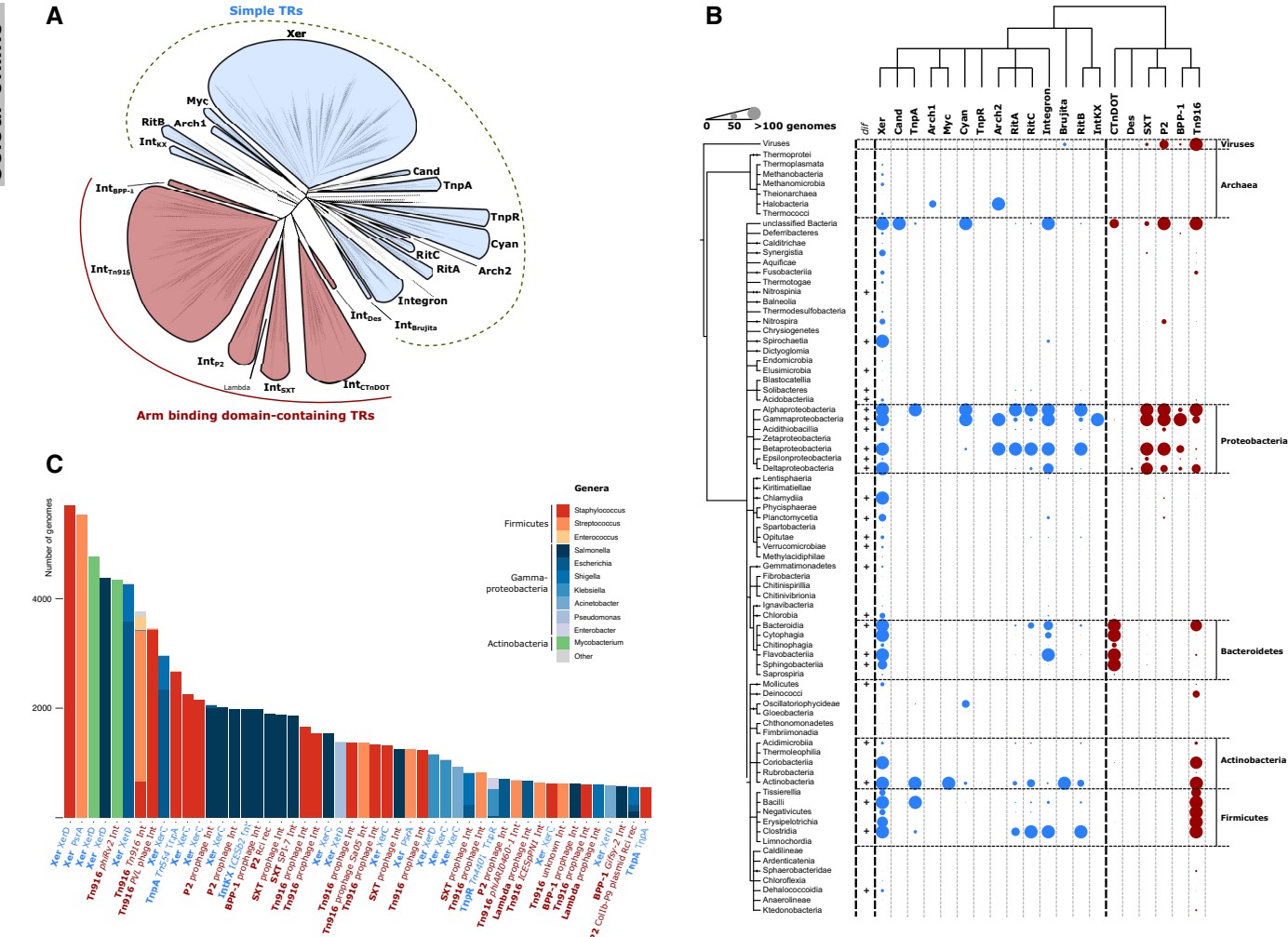

**Figure 1. Diversity and distribution of tyrosine recombinases (YRs).**

A Maximum-likelihood phylogenetic tree of YRs. Two major groups of YRs, simple and arm-binding (AB) domain-containing YRs, are highlighted in blue and red, respectively. YR subgroups are shown as leaves in the tree. Statistical support for branching was evaluated by aBayes, and for all of the subgroups, its value is more than 0.98.

B Taxonomic distribution of YRs. On the top, a schematic tree of the YR phylogeny corresponding to panel (A) is shown (only nodes with statistical support of more than 0.98 are shown). Phylogeny of the bacterial taxa is shown on the left. The abundance of YRs from a specific subgroup in a particular taxon is indicated by different size dots in the plot (colored as in (A)). The exact numbers of genomes are provided in Dataset EV1.

C The fifty most abundant YR proteins found in the genomic sequences available from NCBI. The bars indicate YR abundance in different bacterial taxa with distinct colors. The YRs are named by the subgroup name (in bold) and functional classification. The names of simple and AB domain-containing YRs are colored like in (A). NCBI GI numbers for all the sequences are available in Dataset EV2.

Source data are available online for this figure.

related to the transposase of the Tn554 transposon (Bastos & Murphy, 1988), which have two beta-strands inserted on their N-terminus and an alpha helix insertion between CB and CAT domains (Fig 2 and Appendix Fig S5). The integron-like subgroup contains the integrase proteins of integrons, which act to reshuffle gene cassettes in integron recombination platforms to promote bacterial adaptation (Collis & Hall, 1992). These proteins contain a specific insertion within the CAT domain fold, which is not fully conserved in the subgroup, but is essential for integron function (Fig 2 and Appendix Fig S3) (MacDonald *et al*, 2006). The TnpR subgroup includes the TnpR protein from the carbapenemase-

carrying Tn4401 transposon (Naas *et al*, 2008) and related proteins found in beta- and gammaproteobacteria. Notably, TnpR is one of the most abundant and widely distributed YRs in bacterial genomes, which suggests a prominent role for Tn4401 in propagating carbapenem resistance (Fig 1C). Proteins in the TnpR subgroup have a helical N-terminal DUF3701 domain in addition to the canonical CB and CAT domains (Fig 2 and Appendix Fig S1). The function of this domain is not known, but it contains a helix–hairpin–helix motif, suggesting a role in DNA binding. The Int_Brujita subgroup is formed by YRs of mycobacterial phages. Its best-characterized member is the Brujita phage integrase, which was

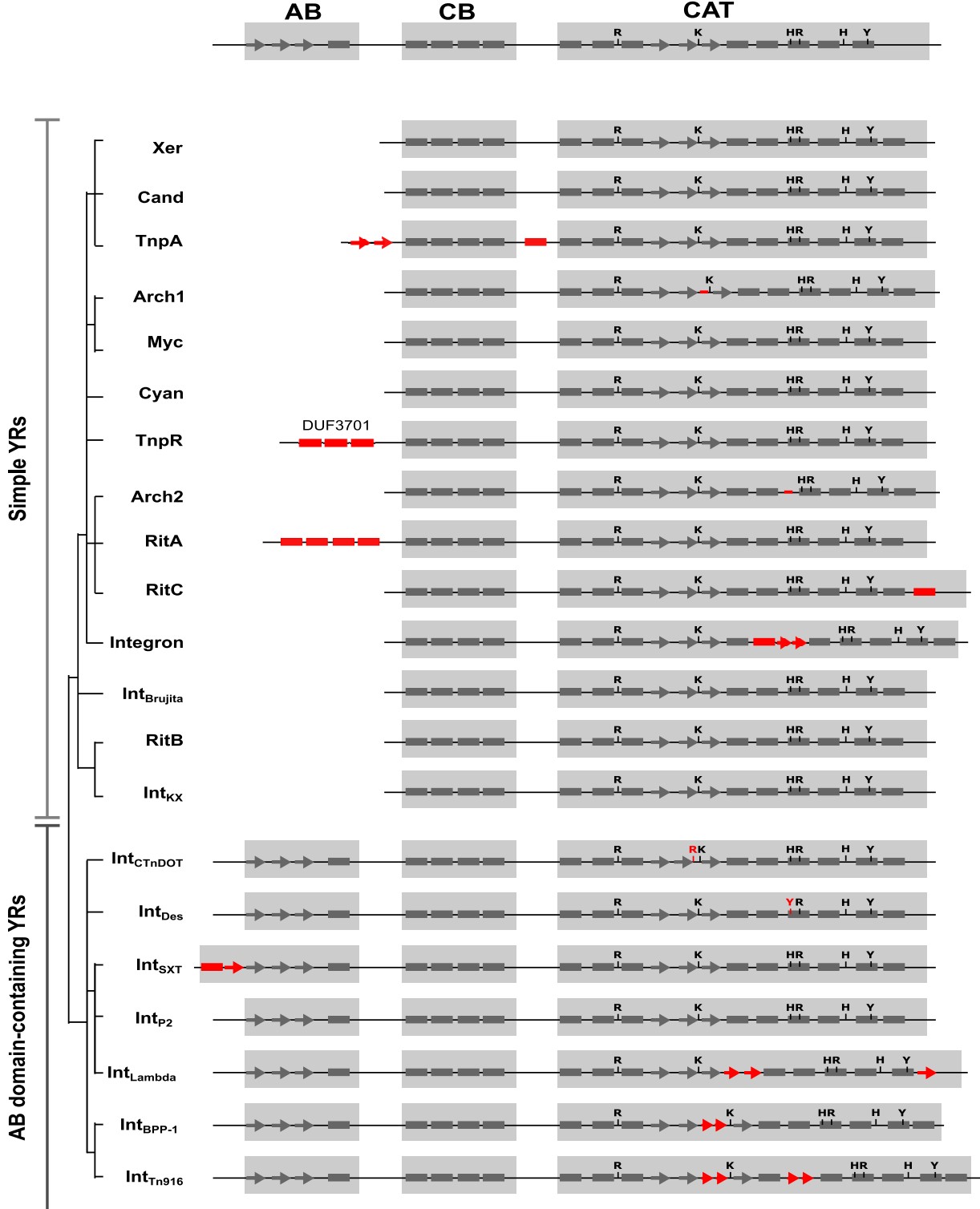

**Figure 2.  Conservation analysis of tyrosine recombinase (YR) subgroups.**

For each of the subgroups, secondary structures of a representative family member were predicted using Jpred or retrieved from corresponding Protein Data Bank (PDB) entries. Helices and strands are shown as rectangles and arrows, respectively. The tyrosine nucleophile and the catalytic RKHRH pentad are marked. Characteristic structural variations of YRs that are conserved within distinct subgroups are highlighted in red. AB—arm-binding domain; CB—core-binding domain; CAT—catalytic domain; DUF3701—domain of unknown function (Pfam accession number—PF12482).

shown to mediate insertion of the phage genome in target bacteria (Lunt & Hatful, 2016). Arch2 subgroup includes proteins related to the integrase of the temperate pleolipovirus SNJ2 from halobacteria (Wang *et al,* 2018). These YRs have a specific short sequence insertion in the CAT domain (Fig 2 and Appendix Fig S3). Finally, the Int$_{KX}$ subgroup (defined in (Roche *et al,* 2010)) comprises YRs found in known pathogenicity islands in human pathogenic gammaproteobacteria (such as PAPI-1, ICESb2, and ICEEc2; Dataset EV3) (Collyn *et al,* 2004; Mohd-Zain *et al,* 2004; Harrison *et al,* 2010; Roche *et al,* 2010; Seth-Smith *et al,* 2012; Juhas *et al,* 2013). Taken together, these analysis results revealed that most of the simple Xer-related YR proteins act in chromosome maintenance, with some distinct, smaller subgroups specifically linked to MGEs. Notably, the latter subgroups are characterized by an increased structural complexity compared to canonical Xer proteins.

### Arm-binding domain-containing tyrosine recombinases

The second large YR group unites proteins that contain, an AB domain in addition to the CB and CAT domains (Appendix Fig S1). The best-characterized members of this group act as integrases of phages or ICEs. This AB domain-containing YR group consists of six major subgroups that are discussed in detail in the following sections.

### Int$_{Tn916}$ subgroup

The largest subgroup of AB domain-containing YRs is the Int$_{Tn916}$ subgroup. It is the most diverse among the AB domain-containing YRs and contains integrases of numerous well-documented ICEs and phages. Its members are most highly represented in gram-positive bacteria, but we also found some examples in other taxa, such as Fusobacteria, Synergista, and Chlamydia (Fig 1B). This subgroup contains some of the most abundant AB domain-containing YRs, such as the mycobacterial phiRV2 prophage integrase (Cole *et al,* 1998) and the integrase of the tetracycline resistance-carrying Tn916 transposon (Franke & Clewell, 1981), each found in the genomes of about 4,000 bacterial strains (Fig 1C).

Generally, members of the subgroup contain an AB domain on their N-terminus, which features three beta-strands and one alpha helix (Figs 2 and 3), as seen in the NMR structure of the Tn916 integrase AB domain (Wojciak *et al,* 1999). In some cases, the AB domain was not directly predicted by Pfam (Appendix Fig S1), but our subsequent sequence analysis revealed that the AB domain is preserved throughout the subgroup (Fig 3). Another characteristic feature of the Int$_{Tn916}$ subgroup is a conserved beta-stranded insertion between the second and third beta-strands in the CAT domain (Fig 2 and Appendix Fig S3). Recent structural and biochemical work on the Tn1549 integrase showed that this protein segment is important for shaping the DNA substrate for recombination (Rubio-Cosials *et al,* 2018).

Notably, the phage- and ICE-related members of this subgroup do not form separate clusters; instead, most clusters contain integrases from both ICEs and phages (Appendix Fig S6). For example, many actinomycete ICE integrases cluster together with the integrases from actinobacterial phages (see cluster pSAM2 in Appendix Fig S6). Interestingly, many YRs within the clusters integrate their respective MGEs at specific genomic sites, with a reoccurring preference for the conserved flanks of essential genes, such as tRNA encoding genes

(Appendix Fig S6). A notable exception is the specific cluster that includes the Tn916 and Tn1549 integrases, which insert into AT-rich regions without a strict sequence specificity (Trieu-Cuot *et al,* 1993; Scott *et al,* 1994; Wang *et al,* 2000; Lambertsen *et al,* 2018). This feature might have contributed to the success of the respective MGEs in spreading to a broad range of bacteria.

### Int$_{BPP-1}$ subgroup

The Int$_{BPP-1}$ is a smaller AB domain-containing YR subgroup, which is closely related to Int$_{Tn916}$. Its members are found in gammaproteobacteria, betaproteobacteria, and phages (Fig 1B). Examples of this subgroup include putative integrases of the Bordetella BPP-1 phage, the Stx2a phage, and the Salmonella Gifsy-2 phage (McClelland *et al,* 2001; Liu *et al,* 2004; Ogura *et al,* 2015), the latter being one of the most abundant proteins in this subgroup (Fig 1C). Int$_{BPP-1}$ YRs feature an AB domain that is annotated as DUF3596 in Pfam (PF12167; Appendix Fig S1) and exhibits a canonical three beta-strand/one helix structure (Fig 3). Similar to Int$_{Tn916}$ members, the Int$_{BPP-1}$ subgroup features a beta-stranded insertion between the second and third beta-strands in the CAT domain fold (Fig 2 and Appendix Fig S3). Members of the family also have weaker conservation of the first histidine in the catalytic RKHRH pentad (Appendix Fig S4).

### Int$_{CTnDOT}$ subgroup

The second largest AB domain-containing YR subgroup is Int$_{CTnDOT}$. It includes proteins from Bacteroidetes (Fig 1B), such as integrases of the ICE CTnDOT and mobilizable element NBU1 (Shoemaker *et al,* 1996; Whittle *et al,* 2002), as well as YRs from the Salmonella genomic island 1 (SG1) (Doublet *et al,* 2005; Douard *et al,* 2010) (Dataset EV3). Initial Pfam annotation suggested that YRs in this subgroup contain only CB and CAT domains, with a substantially larger predicted CB domain than the one found in simple YRs. However, secondary structure predictions previously proposed that the integrase of a prototype CTnDOT element from Bacteroides comprises a canonical AB domain (Kim *et al,* 2010) (Fig 3) and subsequent biochemical experiments confirmed its interaction with subterminal arm DNA sites in the transposon (DiChiara *et al,* 2007; Wood *et al,* 2010). In agreement, our comparative analysis revealed that the N-terminal segment of all Int$_{CTnDOT}$ members consists of two conserved domains: a canonical CB domain and an upstream AB domain (Fig 3 and Appendix Fig S1). Accordingly, we have updated the corresponding Pfam annotation, which is now available in the new version (Pfam 32.0).

Analyzing sequence logos, we further noted that YRs of the Int$_{CTnDOT}$ subgroup show a weaker conservation of the first arginine residue in the otherwise strictly preserved catalytic RKHRH pentad (Box I in Appendix Fig S2) in the CAT domain. Arginine is present in this position in NBU1, NBU2, and Tn4555 integrases, but it is absent in the integrases of CTnDOT, ERL (S), and Tn5520 elements (Cheng *et al,* 2000). Previous biochemical experiments showed that in the CTnDOT integrase, this residue is functionally substituted by another arginine located further downstream in the protein sequence (Kim *et al,* 2010). Consistently, we found that this alternative arginine is conserved in many integrases in the Int$_{CTnDOT}$ subgroup (see conserved R in Int$_{CTnDOT}$ logo in Appendix Fig S3).

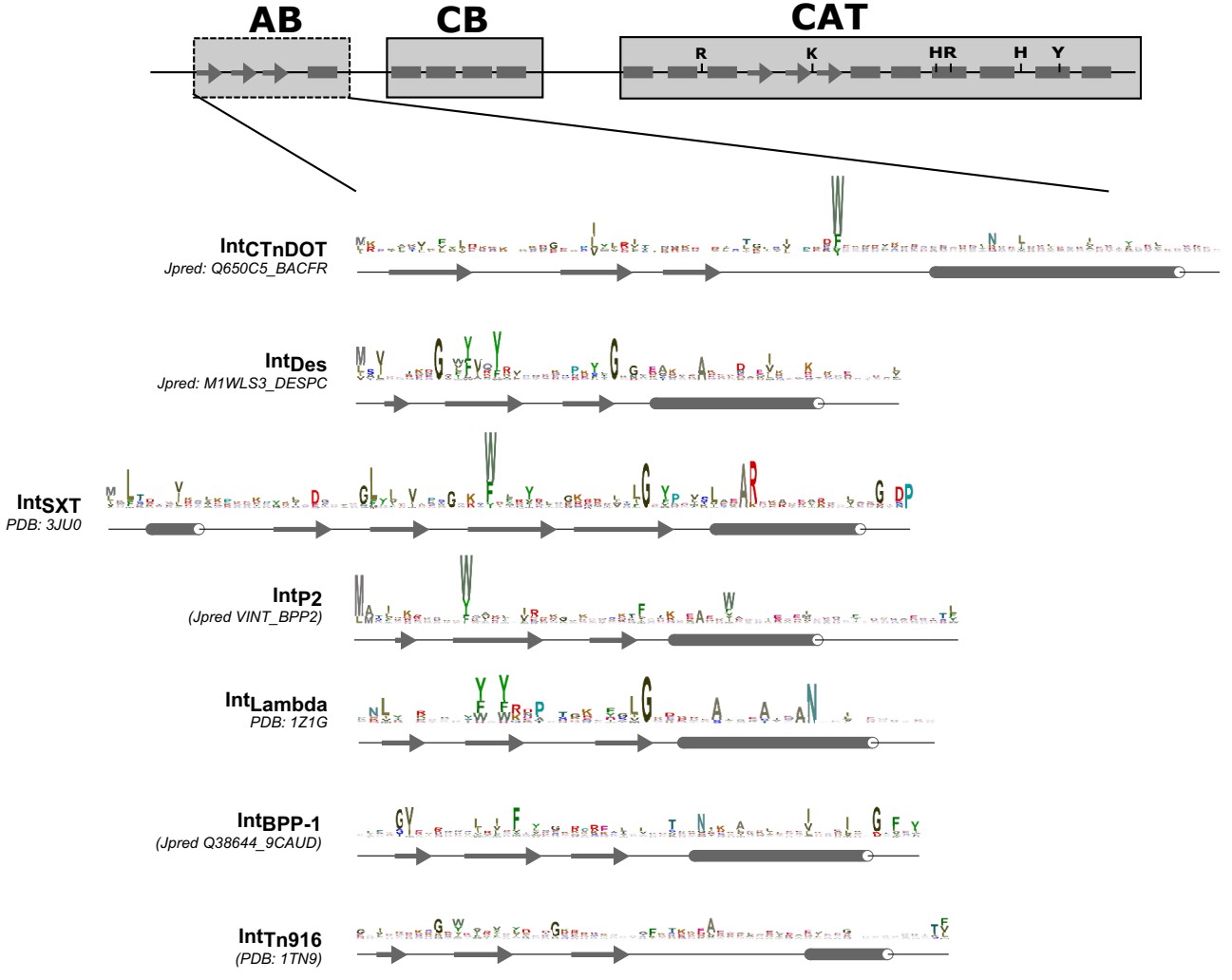

**Figure 3. Sequence conservation of the arm-binding domains of tyrosine recombinases (YRs).**

For each subgroup, web logos were produced after HMM search against the UniProt reference proteomes database and secondary structures were predicted using Jpred or retrieved from corresponding PDB entries (shown below the logos). The logos are colored by residue type, and the typical YR domain composition is shown above the logos as in Fig 2.

Thus, YRs of this subgroup carry the catalytic arginine in one of two alternative locations, resulting in a weaker overall conservation.

### Int<sub>SXT</sub> subgroup

The next large subgroup of AB domain-containing YRs is Int_SXT, which comprises integrases of several ICEs, genomic islands, and phages. A characteristic feature of this subgroup is the presence of an N-terminal DUF4102 domain (Appendix Fig S1). This was previously annotated as an AB domain of genomic island integrases (Szwagierczak *et al*, 2009) and contains an additional beta-strand and an alpha helix compared with AB domains of other YRs (Figs 2 and 3). Phylogenetic analysis revealed that two out of six clusters within the Int_SXT subgroup contain integrases from both ICEs and phages (Appendix Fig S7). Members of major clusters share distinct genomic insertion profiles, integrating their MGEs near essential genes. For example, integrases of the P4 and Sf6 phages cluster

together with various ICE YRs, all of which insert downstream of tRNA genes (P4 cluster, Appendix Fig S7) (Boyd *et al*, 2009; Van Houdt *et al*, 2012). Similarly, integrases of the epsilon15 phage, the CMGI-3 element, and related elements form a separate cluster, and all target the 3′ flank of the *guaA* gene involved in GMP biosynthesis (Kropinski *et al*, 2007; Bi *et al*, 2012) (epsilon15 cluster, Appendix Fig S7). The same pattern is seen for integrases of the Enterobacterial cdt1 phage, the SXT element, and closely related ICEs, all of which insert next to the *prfC* gene encoding a factor involved in termination of translation (Hochhut & Waldor, 1999; Asakura *et al*, 2007) (SXT cluster; Appendix Fig S7). Thus, members of each Int_SXT cluster seem to drive their diverse MGEs into specific locations, perhaps owing to characteristic features in the integrase sequences. Their preference for the flanks of conserved genes might help promote their dissemination between species and explain their characteristic taxonomic distribution. In addition, the mixed distribution of ICE and phage integrases suggests that these elements

frequently exchange their integrases. This is also supported by previous observations that ICEs with different conjugation machineries have closely related integrases (Cury *et al*, 2017).

### Int<sub>P2</sub> subgroup

The Int$_{P2}$ subgroup of AB domain-containing YRs contains integrases from proteobacterial phages, such as HP1 and P2. Another interesting member of this subgroup is the plasmid-borne Rci recombinase, which regulates R64 plasmid conjugation by reshuffling distinct gene segments to generate diverse pili proteins (Komano *et al*, 1987; Gyohda & Komano, 2000; Roche *et al*, 2010). The CAT domains of YRs in this subgroup are highly similar to those of simple YRs, as also seen with previously determined crystal structures (Hickman *et al*, 1997; Skaar *et al*, 2015). Most YRs in this subgroup contain an AB domain with a classical fold (Fig 3), except the Rci recombinases that lack the AB domain. In agreement with previous sequence analyses (Boyd *et al*, 2009), our phylogenetic reconstructions suggest that Int$_{P2}$ YRs are related to the lambda phage integrase; however, this clustering is not well supported by statistical analysis (Fig 1A and Appendix Fig S1). Although the well-studied lambda phage integrase is often used as a prototype for the tyrosine recombinase superfamily (Landy, 2015), our analysis revealed that it is quite different from other YRs. It contains substantial alterations even in the CAT domain, including an insertion of two beta-strands after the third beta-strand of the canonical fold, and the replacement of the C-terminal alpha helix with a beta-strand (Fig 2, Appendix Figs S3 and S4).

### Int<sub>Des</sub> subgroup

Finally, Int$_{Des}$ is a small subgroup of AB domain-carrying YRs. Its members are found only in the genus Desulfovibrio of Deltaproteobacteria (Fig 1B). This subgroup features specific sequence perturbations in the catalytic core: Namely the first arginine residue of the RKHRH pentad is shifted in comparison with other YRs and the first histidine is substituted with a tyrosine (Appendix Figs S2 and S4). The biological function of these YRs has remained unknown to date.

### Identification and classification of integrative and conjugative elements

The vast majority of the YRs that we analyzed remain unannotated in genomic databases. This particularly hinders identification and characterization of YR-carrying MGEs. To test whether our classification system can help predict YR function, we next checked whether the unannotated YRs found in ICE-related subgroups are indeed integrases of ICEs. For this, we examined the YRs' genomic neighborhood to identify known conjugative machinery proteins (as in Guglielmini *et al*, 2014; Abby *et al*, 2016). If an integrase was found in proximity (± 100 kb) to known conjugation machinery proteins, then the corresponding region was considered to be a putative ICE (Fig 4A). ICEs retrieved from the ICEberg database were used for benchmarking. This analysis revealed a total of 59 previously unannotated ICEs (Appendix Fig S8, Dataset EV4). The putative ICEs were then further validated by manual identification of their terminal repeat sequences. We confidently identified terminal repeats in 50 out of 59 predicted ICEs. For 49 of these, the

conjugation machinery was found within the predicted borders of the element, further confirming their identity. In one predicted element, the conjugation machinery was located outside of the borders (Dataset EV4), suggesting a coincidental co-occurrence of YR and conjugation genes in this instance.

To further characterize the detected ICEs, we aimed to reconstruct the naive insertion site (i.e., the bacterial genomic sequence prior to integration) of the identified ICEs and look for such undisrupted sites in closely related genomes. As functional ICEs can move to new genomic sites, successful identification of naive sites can provide ultimate confirmation of their identity and mobile nature. However, identification of such naive sites requires recent mobility of the ICE and may also be challenged by a limited availability of complete genome sequence data for related species in public databases. Nevertheless, we found naive sites for 18 out of the 49 ICEs, which further validates these elements and indicates their recent activity (Dataset EV4, Appendix Fig S9).

YRs in the new ICEs belonged to five YR subgroups (Fig 4B, Dataset EV4), with most examples found in the Int$_{Tn916}$ (23), Int$_{P2}$ (17) and Int$_{SXT}$ (14) subgroups. To further analyze the detected ICEs, we next reconstructed the phylogeny of their YRs and plotted the genetic structure of their respective conjugation machineries (Fig 4B and Appendix Fig S8). ICEs with closely related YRs were generally associated with closely related conjugation systems, but ICE groups with somewhat more distantly related YR proteins often contained unrelated types of conjugation modules (Fig 4B and Appendix Fig S8). For instance, ICE groups that carry YRs from the diverse Int$_{Tn916}$ and Int$_{SXT}$ subgroups revealed various conjugation systems. In turn, some clusters of the Int$_{SXT}$ YRs and the distinct Int$_{KX}$ YRs associated with the same conjugation system, called MPF$_G$ (Fig 4B and Appendix Fig S8), located on different sides of the YR. Altogether, this suggests recurrent exchange of conjugation modules between distantly related ICEs, in accordance with previous reports (Cury *et al*, 2017).

Furthermore, to complete the characterization of the ICEs' mobilization machinery we looked for excisionase (Xis) genes within newly identified and previously reported ICEs (Fig 4B and Appendix Fig S8). Xis regulates the directionality of the recombination reaction in some of the known YR-containing systems (Connolly *et al*, 2002; Wood & Gardner, 2015). We found that only AB-containing YRs are associated with Xis proteins, which may suggest potential cooperation between the AB domain and Xis. Consistent with this idea, a physical interaction was recently proposed for the integrase and Xis of the lambda phage (Cho *et al*, 2002; Laxmikanthan *et al*, 2016). We could not detect Xis in any of the 15 ICEs with simple YRs from the Int$_{KX}$ subgroup.

Taken together, successful identification of new ICEs confirms the predictive value of our classification system for automated annotation of YR function and demonstrates its utility to improve characterization of the bacterial mobilome.

## Discussion

### Xer tyrosine recombinases are conserved and ancient

In the present study, we devise a classification system for bacterial YRs that are related to Xer recombinases. Based on phylogenetic

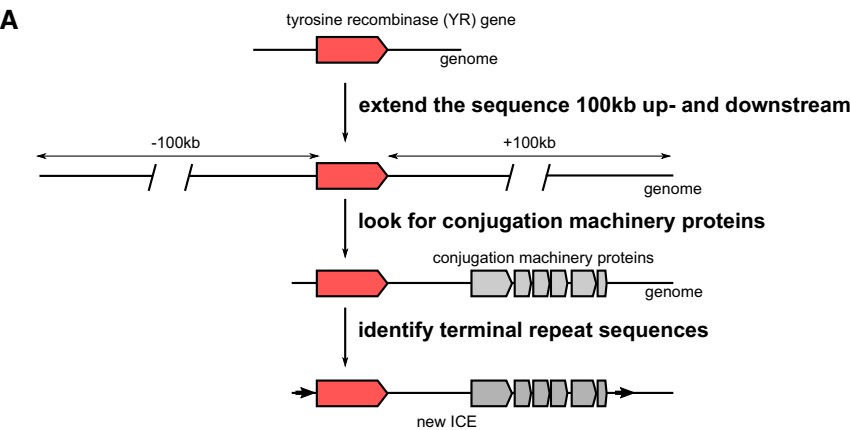

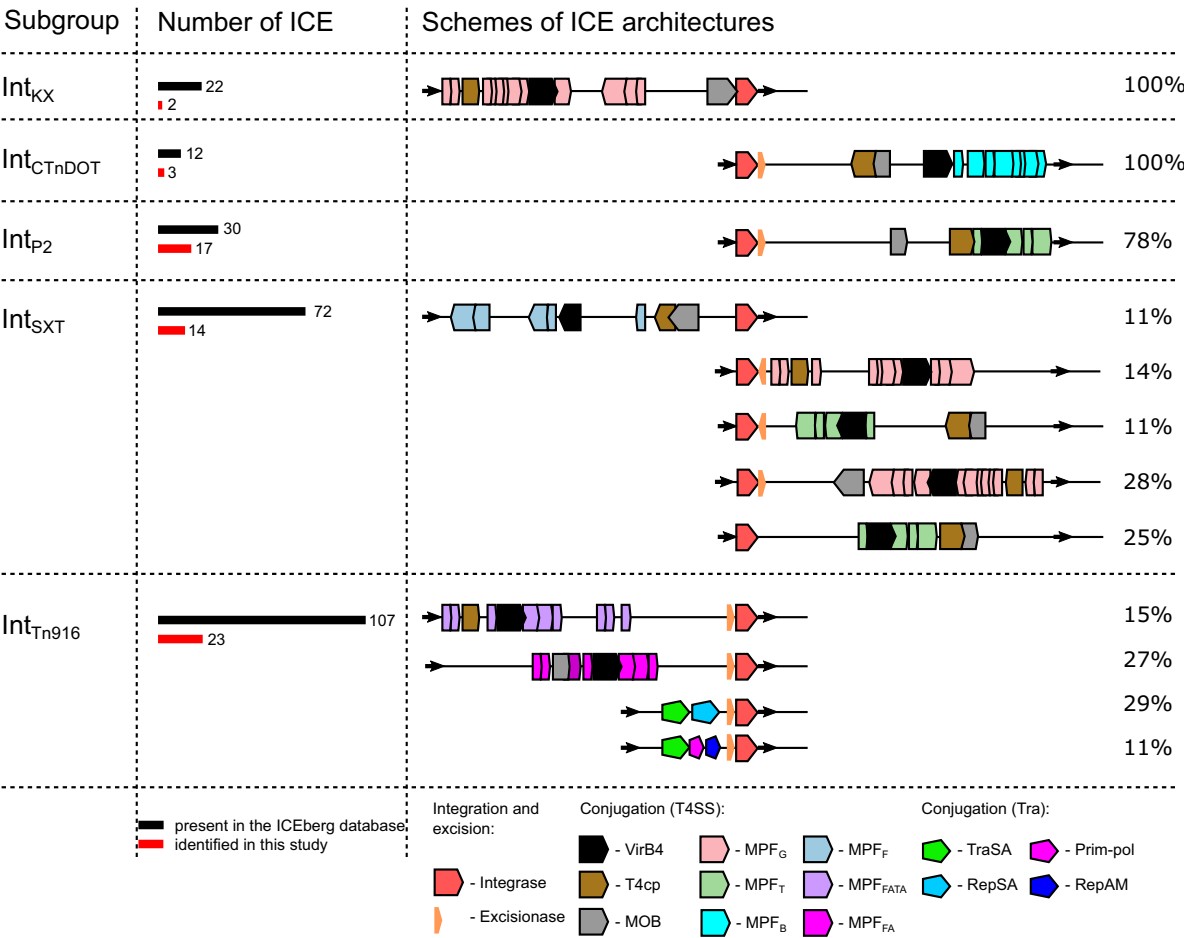

**Figure 4. Tyrosine recombinase-based ICE identification and characterization.**

A   Overview of the computational pipeline for ICE identification. The genomic regions of the tyrosine recombinase (YR) genes were expanded 100 kb upstream and downstream and analyzed for the presence of conjugation-related genes and repeat sequences.

B   Structural diversity of YR-carrying ICEs. All ICEs clustered into five subgroups based on their YR classification (left). The numbers of ICEs in each of the subgroups are displayed as bars with numbers (middle). Schematic representations of ICE architectures are shown, aligned by their integrase genes (red symbol, right). Protein open reading frames of various types of conjugation machineries are depicted with different colors as indicated at the bottom of the figure.

 

reconstruction, we divided all YR proteins into two groups and twenty subgroups. Of these, the most abundant and widely distributed is the Xer subgroup of simple YRs, which includes close homologues of the chromosome dimer resolution proteins XerC/D from 16 bacterial phyla (Fig 1 and Appendix Fig S1). Notably, recent reports described an identical taxonomic distribution of conserved *dif* DNA sequences (Kono *et al,* 2011), which serve as Xer recombination sites on bacterial chromosomes, thus implying a widely conserved functional role for these proteins. The phylogeny of XerC/D recombinases in proteobacteria was further found to be correlated with their host taxonomy, suggesting that their evolution follows a strictly vertical trajectory (Carnoy & Roten, 2009). This wide distribution and vertical inheritance of Xer-*dif* systems indicate that Xer proteins are the most ancient type of bacterial YRs, which may have served as the evolutionary source for other more complex YRs.

## Simple tyrosine recombinases can drive the movement of mobile genetic elements

Besides the chromosomal Xer proteins, we identified several simple YRs encoded on MGEs, such as phages or ICEs. Most of these YRs belonged to two subgroups, the Int$_{Brujita}$ subgroup found in mycophages and the Int$_{KX}$ subgroup from gammaproteobacterial ICEs; a few isolated examples fell into the Xer and RitB subgroups (Datasets EV1 and EV3). Notably, several of these proteins have been shown to be required for mobilization of their respective elements (Qiu *et al,* 2006; Fischer *et al,* 2010; Flannery *et al,* 2011; Lunt & Hatfull, 2016). For example, the simple YR of the Brujita phage actively drives its excision and integration (Lunt & Hatfull, 2016), and XerT from Helicobacter pathogenicity islands is needed for their horizontal transfer (Fischer *et al,* 2010). The close relationship of these MGE-borne YRs with the Xer subgroup and their significance for MGE function suggest that phages and ICEs have repeatedly sequestered Xer genes from their host genomes and functionally repurposed these to drive their transfer. Alternatively, MGE integrases may have been domesticated by the host for dimer resolution functions, as proposed recently (Koonin *et al,* 2020). However, this contradicts the observed vertical evolution and wide taxonomical distribution of Xer genes.

## Acquisition of Arm-binding domain drove the evolution of mobile genetic element integrases

While some phages and ICEs carry simple YRs, most of them possess more complex AB domain-containing YRs. In fact, our analysis detected 27 phages with simple YRs versus 410 with AB domain-carrying YRs (Dataset EV1). Similarly, we found 12 ICEs that carry only simple YRs, whereas 211 had AB domain-containing YRs (Dataset EV3). These complex YRs cluster into a monophyletic group with six distinct subgroups, where all subgroups contain an N-terminal AB domain with similar secondary structure (Fig 3), suggesting a common evolutionary origin. Although the AB domain is less conserved than the CAT domain, sequence similarity can be detected between different YR subgroups. Pfam AB domains from all subgroups belong to the same clan (CL0081), and sequence logos show significant conservation within and across subgroups (Fig 3). In agreement, previous structural studies also revealed a similar AB

fold in the lambda phage integrase and in Int$_{Tn916}$ and Int$_{SXT}$ family YRs (Clubb *et al,* 1999; Wojciak *et al,* 2002; Szwagierczak *et al,* 2009).

The functional role of the AB domain may be inferred from its function in the lambda phage integrase, one of the best-characterized members of this YR group. Here, the AB domain binds to internal arm DNA sequences within the phage genome and plays an essential role in guiding DNA recombination toward excision or integration in different stages of the phage lifecycle (Tirumalai *et al,* 1996; Biswas *et al,* 2005; Radman-Livaja *et al,* 2006). The presence of an AB domain in the vast majority of ICE and phage integrases indicates that these elements generally benefit from the function of this domain in regulating their integration and excision. MGEs that carry simple YRs may represent an earlier step in evolution with less intricate regulatory features (Lunt & Hatfull, 2016). Interestingly, large serine recombinases that perform integration and excision of discrete phages also require a separate DNA-binding domain for precise regulation (Rutherford & Van Duyne, 2014), indicating that acquisition of accessory DNA-binding domains may be a common strategy in MGE evolution.

## Tyrosine recombinase classification aids the annotation of mobile genetic elements in bacterial genomes

The recent increase in bacterial genome sequence data highlighted the impact of MGEs and motivated the development of automated sequence-based MGE mining tools. For insertion sequences, the simplest MGEs that typically contain an RNaseH-like DDE transposase flanked by short inverted repeats, existing pipelines provide confident annotation through homology-based prediction (Xie & Tang, 2017). The recently communicated TIGER pipeline also maps various integrative genomic elements by using Pfam-based annotations (Mageeney *et al,* 2020). However, for YR-carrying elements the close relationship of YR family transposases/integrases to essential bacterial genes has greatly hampered functional annotation. In particular, the TIGER software tackles this by discarding Xer and Integron-related sequences, assuming that all other YRs are MGE integrases, which results in false-positive hits (Mageeney *et al,* 2020). Through comprehensive YR classification, we found that the presence of an AB domain in a YR is a strong predictor of its function in phage or ICE mobility. In fact, acquisition of this AB domain by YRs and their cooperation with Xis proteins may have driven evolutionary adaptation of YRs to a "mobile lifestyle", helping to promote and regulate MGE movement in and between bacterial cells. Sequence features in all YR domains further support functional annotation of MGE-borne proteins and can even enable identification of simple YRs with a specific mobility function. These direct sequence-to-function relationships open up new opportunities for functional annotation and provide clear rules for predicting the mobile nature of YRs and the genomic regions they reside in.

By using our classification system, we identified new ICEs in diverse genomes with mobile YRs as markers. We found 59 previously unannotated ICEs, substantially expanding the repertoire of known elements and illustrating the power of our approach for automated MGE detection. Going forward, application of our pipeline will help explore the abundance and diversity of MGEs in bacterial genomes in diverse environments. Comprehensive studies of

 

MGE content, distribution, and dynamics are necessary to track MGE transfer and to assess their impact on genetic exchange and bacterial adaptation. These studies are important to better understand the dynamics of microbial communities and to follow the spread of genetic traits, such as antibiotic resistance. Our work prepares the stage for analyzing the full repertoire of MGEs and their cargo in bacterial genomes, thus offering new opportunities to characterize gene flow in bacterial communities.

# Materials and Methods

**Reagents and Tools table**

| Reagent/Resource | Reference or source | Identifier or catalog number |
| --- | --- | --- |
| **Software** | | |
| PhyML | Guindon *et al*, 2010 | 3.0 |
| Jackhmmer | https://www.ebi.ac.uk/Tools/hmmer/search/jackhmmer | 2016_01 |
| Hmmer | http://hmmer.org/ | 3.1b2 |
| Skylign | http://skylign.org/ | – |
| CD-HIT | http://weizhongli-lab.org/cdhit_suite/cgi-bin/index.cgi | – |
| MAFFT | Katoh & Standley, 2013 | Version 7 |
| iTol | https://itol.embl.de/ | V2 |
| Mauve | Darling *et al*, 2010 | Snapshot_2015-02-25 |

**Methods and Protocols**

*Identification and phylogenetic analysis of tyrosine recombinases from bacterial genomes*

As the first step, we used the XerD amino acid sequence (GenBank accession number CDO12527) as a query in a jackhmmer iterative search against the rp15 (v.2019_08) database with different e-values (from 1e-05 to 1e-40). Our aim was to include as many remote Xer homologues as possible, which is possible through the jackhmmer procedure (Park *et al*, 1998), but at the same time avoid inclusion of distantly related sequences, such as eukaryotic YRs from DIRS retrotransposons, that hamper reconstruction of the phylogenetic tree. We chose to use e-value of 1e-30 as this threshold discards 96% of eukaryotic YRs (from those included in Pfam family) and retains more than 80% of bacterial YRs (Appendix Table S2) (Goodwin & Poulter, 2001; Van Houdt *et al*, 2012). We then repeated the same analysis with this e-value but now against the full UniProt reference proteomes database (converged after 37 iterations; accessed 13.01.2016, http://hmmer.janelia.org/search/jackhmmer currently moved to https://www.ebi.ac.uk/Tools/hmmer/search/jackhmmer; (Finn *et al*, 2011)). Other well-known YRs, including diverse bacterial or phage YRs for which crystal structures, are available (XerD, XerH, integrases of phages λ, HP1 and P2 and integron integrase) and most ICE-encoded YRs from the ICEberg database passed this threshold and therefore were included in the study. 187 out of 191 (~98%) YRs of single YR-containing ICEs in ICEberg also passed the threshold and were annotated in our study (Dataset EV3). At the next step, all the sequences that passed the threshold (altogether 9,909 proteins) were then clustered with a 40% identity threshold using CD-HIT (Fu *et al*, 2012) and the representatives of clusters with more than one sequence were used for further analysis. This was implemented to exclude singletons and rare sequences with truncations. Then, altogether 866 sequences were aligned using the E-INS-I method from the MAFFT software, recommended for sequences where several conserved motifs are embedded in long unalignable regions (Katoh & Standley, 2013). In the alignment, columns containing more than 80% of gaps and the N-terminal region of the alignment (corresponding to AB) were removed. The final alignment is available in Source Data. The phylogenetic tree was constructed using the PhyML package (Guindon *et al*, 2010) with LG+I + G model of protein evolution and evaluated by ProtTest (Abascal *et al*, 2005). In the course of the reconstruction, we built 1,000 trees using both NNI and SPR moves as topology search and random tree as starting trees. A tree with the highest maximum-likelihood value was used as the reconstruction of the YR phylogeny (Fig 1A). The branch support was evaluated with aBayes and non-parametric SH-aLRT (Shimodaira & Hasegawa, 1999; Anisimova *et al*, 2011). We further defined subgroups as clades that (i) include more than four sequences, (ii) have aBayes and SH-aLRT branch support values of more than 0.98 and 0.85, respectively, and (iii) exhibit similar domain composition of its members (i.e., presence/absence of distinct Pfam domains (Finn *et al*, 2016)). The tree with the domain composition retrieved for each of the sequences is available as Appendix Fig S1. The same tree with all branch supports is available at http://itol.embl.de/shared/gera for interactive inspection.

The phylogenetic trees for $Int_{Tn916}$ and $Int_{SXT}$ subgroups (Appendix Figs S6 and S7, respectively), as well as for ICE YRs from Appendix Fig S8, were produced in the same way as for the general tyrosine recombinases tree and their branch support was evaluated by aBayes. The sequences were mined from the ICEberg database (Bi *et al*, 2012) and the UniProtKB phage section, using profile HMMs specific for the subgroups (see section below). For Appendix Fig S8, the sequences of the newly identified ICEs and ICEs from ICEberg database were used for the reconstruction (Datasets EV3 and EV4).

### Annotation of tyrosine recombinase putative function, domain organization, conservation, and host organism

In order to annotate all YRs from the UniProt reference proteomes, we created profile HMMs for each of the subgroups. For that, we retrieved the complete sequences of the proteins forming the group and reran the MAFFT alignment of the proteins with the same settings as described above. Then, we built a profile HMM for this alignment using HMMER3 (Eddy, 1998) and set the gathering threshold for the profile as the lowest score that was produced by the sequences used to build the group. The collection of the 20 profile HMMs specific to each subgroup is available in Source Data.

To analyze YR distribution, we mapped all proteins that were retrieved in the UniProt reference proteomes search to their genomic locations and built a plot of the distribution of YRs from each subgroup in bacteria, archaea, and viruses (Fig 1B, Dataset EV1), visualized using the iTOL webserver (Letunic & Bork, 2011). In addition, to shed some more light on the abundance of the YRs in bacterial genomes we extracted fifty of the most abundant YRs found in the sequenced genomes and annotated their distribution in bacteria, classification, and putative function (Fig 1C, Dataset EV2).

To visualize the similarities and differences between YRs from the different groups and subgroups, we created logos representing the sequences from each of the subgroups (Appendix Figs S2–S5). For that we used the protein alignments, containing all hits produced by the profile HMM-based search against the UniProt reference proteomes database in the previous step, and visualize them using the Skylign webserver (Wheeler *et al*, 2014). The logo for the Lambda integrase-related YRs, which failed to form a subgroup as they have only two members of the clade in the tree (Appendix Fig S1), was created with only two sequences used for the corresponding profile HMM reconstruction. Secondary structure predictions were done using Jpred 4 (Drozdetskiy *et al*, 2015).

To retrieve functional annotation of the proteins, we run the profile HMM against databases of the proteins with known functions. These included Protein Data Bank (PDB, www.rcsb.org; (Berman *et al*, 2000)), ICEberg (Dataset EV3; http://db-mml.sjtu.edu.cn/cgi-bin/ICEberg/; (Bi *et al*, 2012)), and ACLAME (http://aclame.ulb.ac.be; (Leplae *et al*, 2010)). In addition, the homologues of the identified YRs were detected using BLAST searches available from ISFinder (database on the insertion sequences, www-is.biotoul.fr; (Siguier *et al*, 2012)), INTEGRALL (integron database, http://integrall.bio.ua.pt, (Moura *et al*, 2009)), and NCBI CDD (Marchler-Bauer *et al*, 2017).

### Annotation of new putative ICEs

To predict new ICEs, we created a Perl script NeighborsScan, which is available from GitHub repository (https://github.com/smyshlya/NeighborsScan). This script uses protein accession numbers as an input, retrieves their genomic position, expands this position 100 Kb upstreams and downstream, downloads all proteins from this region, and runs hmmsearch against these proteins using user provided collection of profile HMMs (such as conjugation machinery protein profile HMMs). The results of this run can be directly visualized using iTol web server (Letunic & Bork, 2011). Although some ICEs may be up to 600 Kb in length, we chose to cover only ± 100 kb region as a good compromise between the sensitivity of the search and the computational time. The conjugation machinery proteins were identified using hmmscan search against profile HMMs available from CONJdb with the default e-value of 1e-3 (Guglielmini *et al*, 2014; Abby *et al*, 2016). For excisionase protein identification, we used Pfam profile HMMs HTH_17, HTH_31, Phage_AlpA, MerR, and DUF2384 with corresponding gathering thresholds. Genomic regions that contain integrase and conjugation machinery proteins were predicted as ICEs. All of the full-length ICEs present in the ICEberg database were confirmed using this approach, providing a benchmark for the method.

To further validate the identification of new ICEs, we used three levels of computational verification as follows: First, we predicted the borders of the identified ICEs. In known ICEs, terminal repeats define the ends of the element, which are recognized by the integrase enzyme to execute ICE excision and integration. Typically, the integrase gene is located adjacent to one of these terminal repeats. Thus, identification of such terminal repeats and verification of their location relative to the integrase gene will verify the identity of our newly annotated ICEs. These repeats are reported in the Dataset EV4.

Second, we tested whether a complete conjugation system is encoded within the predicted borders of the candidate ICEs and shows sequence conservation in closely related elements. Active ICEs generally include conjugation machinery genes, which "travel" with the ICEs and are required for their intercellular transfer. The requirement for these modules implies a certain level of conservation within ICE families. Our analysis revealed that closely related ICEs have closely related conjugation systems (Fig 4 and Appendix Fig S8).

Third, we attempted to identify naive (ICE-free) insertion sites for confirmed ICEs in closely related genomes (Appendix Fig S9, visualized with Mauve software Darling *et al*, 2010). Successful identification of such naive sites provides direct confirmation of ICE movement but may be compromised by the limited availability of related genomes in public databases.

All ICEs identified in the study are presented in Dataset EV4 and were submitted to the ICEberg database.

## Data availability

The computer code produced in this study is available in the following GitHub repository: https://github.com/smyshlya/NeighborsScan.

**Expanded View** for this article is available online.

### Acknowledgements

The authors thank Michael Chandler and Cecilia Zuliani for reading and commenting on the manuscript, and members of the Bateman and the Barabas labs for helpful discussions. This work was supported by the EMBL, the Federal Ministry of Education and Research of Germany (BMBF) under JPIAMR 2016 (01KI1706, JumpAR) and the EMBL Interdisciplinary Postdoc (EIPOD) program under Marie-Curie COFUND (291772 G.S.). Open Access funding enabled and organized by Projekt DEAL.

### Author contributions

GS performed the study. GS, AB, and OB designed the study and wrote the manuscript. All authors read and approved the final manuscript.

### Conflict of interest

The authors declare that they have no conflict of interest.

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
