## [Review Process File · Molecular Systems Biology]

Sequence analysis of tyrosine recombinases allows annotation of mobile elements in bacterial genomes

Georgy Smyshlyaev, Alex Bateman, and Orsolya Barabas

DOI: [10.15252/msb.20209880](https://doi.org/10.15252/msb.20209880)

Corresponding author(s): Georgy Smyshlyaev (g.smyshl@gmail.com)

Review Timeline:

Submission Date:	4th Aug 20
Editorial Decision:	26th Aug 20
Revision Received:	23rd Mar 21
Editorial Decision:	8th Apr 21
Revision Received:	18th Apr 21
Accepted:	20th Apr 21

Editor: Maria Polychronidou

Transaction Report:

Thank you again for submitting your work to Molecular Systems Biology. We have now heard back from the three referees who agreed to evaluate your study. As you will see below, the reviewers acknowledge that the presented findings seem timely and relevant for the field. However, they raise a series of concerns, which we would ask you to address in a major revision.

I think that the recommendations of the reviewers are rather clear and there is therefore no need to repeat the points listed below. One of the more substantial issues is raised by reviewer #2 and refers to the need to provide some level of validation for the newly annotated ICEs. All other issues raised by the referees would need to be satisfactorily addressed. Please let me know in case you would like to discuss in further detail any of the issues raised.

On a more editorial level, we would ask you to address the following issues.

REFEREE REPORTS

Reviewer #1:

In this manuscript, the authors investigate classification of tyrosine recombinases in bacterial genomes. They provide a thorough analysis which is important, timely and interesting even for those not deeply involved in bacterial genomics. It's also written in a clear and accessible way, suitable for a general audience who may not be familiar with all of the methodological issues. I do have a few comments on the manuscript which I hope will improve it if attended to.

Major comments

The classification of the recombinases into subgroups, as illustrated in Figure 1A, looks very robust at least at the level of the major groups. Nevertheless, it is a slight concern here that the authors have picked one method and base all their conclusions on this. Alternatives should be tried to validate the robustness of the classification. Specifically

1) Can the authors investigate using simple blast as an alternative to jackhammer?

2) The e-value cutoff is not explained. It seems to strike a balance between sensitivity and specificity but I think it would be a sound plan if the authors were to show that their classification is robust and that their cutoff optimises sensitivity and specificity. It would thus be good if the authors tried i) a more restrictive cutoff ii) a much more lenient cutoff (say, $1e-5$)

3) Similarly, the hmmscan cutoff was $1e-20$, which is not consistent with the $1e-30$ cutoff used for the jackhammer- no explanation was given. Some investigation of the robustness of the results to this parameter would also be helpful.

The identification of conjugation machinery in the methods is described as being done by a "custom perl script" which is available through GitHub. It's great that the script is available but it would be important for the authors to also explain what the script is doing and how it is working in the methods section- it was not obvious to me how these proteins are identified and whether this relies on previous classification or not.

Minor comments

For future submissions, it might be worth including line numbers for ease of reference for reviewers.

Page 3

"Some MGEs hijack host-encoded Xer proteins[8,9] while others encode specific TRs to promote their own integration..."

Which part of the sentence does the reference refer to? Please add references if required for the second part of the sentence

"TRs contribute to the emergence of antibiotic resistance on several ways..."
Should read "in several ways"

P4

"Building on the largely expanded..."

Not very clear. Might as well just remove the first subclause and instead begin the sentence:

"Here we assembled an extended..."

"To comprehensively analyse"

Change to "To analyse" - can't be sure that you've covered everything using this method [Or "to boldly analyse" if you really want a split infinitive here]

"jackhammer search"

If you are to mention jackhammer in the main text I think you should give a brief explanation of what this is, and also a reference to the hmmer version where jackhammer was first introduced, similarly to the way you describe HMM on p5.

P5

"Two major phylogenetic groups..."

The figure looks very convincing but it would be worth explicitly stating the branch support values for these two groups in the text. Similarly, what are the branch support values for the subgroups in the following descriptions?

P12

"we divide all TR proteins into two groups..." should be "we divided.."
"been experimentally shown" should be "been shown"

Reviewer #2:

Summary:

Smyshlyaev et al. begin their sequence analysis of Tyrosine Recombinases (TRs) by searching the proteome database using the XerD protein from *E. coli*. Then they generate a phylogenetic tree and utilize HMM to find all TR homologues present in UniProt proteomes collection. A preceding maximum likelihood phylogenetic tree analysis for representative sequences from diverse tyrosine recombinase families was carried out by Wang J. et al. in 2018 (published in NAR). However, a further investigation of Xer recombinases was required, since it is clear that they have been recruited from MGEs. Here in this manuscript, Smyshlyaev et al. pave the way to the understanding of TRs with a more comprehensive analysis. The authors then extract the 50 most abundant TR proteins and characterize them further. This analysis reveals that all TRs can be classified in two major phylogenetic groups based on the presence or absence of the highly variable N-terminal Arm-Binding (AB) domain. Following this, the authors provide a detailed depiction of several sub-groups within these two major groups. These sections are descriptive sequence analysis of each subgroup of TR homologues and as well include the speculations on their function. Nevertheless, the focus on functional and conserved residues among the TRs within each subgroup leads to the novel insights. For instance, the commonly used prototype; lambda phage integrase is apparently quite different from other TRs. Another novel finding is the high correlation of the presence of AB domain and the mobile character of the TR, as a part of Integrative and Conjugative Elements (ICEs) or as phage integrases. This observation was followed up with the identification of new, putative ICEs by the presence of AB-containing TR in the genomic vicinity of conjugative elements. The method was bench-marked on ICEberg database and proved to predict all full-length ICEs found in the database as well as 59 previously unannotated ICEs.

General Remarks:

The paper certainly presents a significant body of work, the tools used are solid and generally used properly. The conclusions drawn from the results sound convincing but leave some questions unanswered or not addressed. Overall, this study provides a thorough cataloguing of TRs which is potentially important for MGE identification and their impact on adaptation and will potentially serve as a resource for other interested scientists in the future. The main novelty is linking the presence of Arm Binding domain with the function in ICE or phage mobility and using it as a predictor for ICE annotation. The work presents an interesting basis for further development of sequence-based predictors of TR function.

There is a recent review by Koonin et al. (published in *Nat. Rev. Genetics* 21, pages119-131(2020)) on MGEs and to my surprise Smyshlyaev et al. did not cite this paper. If there was a detailed and broader outlook in addition to the discussion of this study following the points which were raised in Koonin's review, the significance of this study could be pronounced sufficiently.

Major points:

1. The most conspicuous point of this work lies in the phylogenetic analysis of TR dataset. The

presented phylogeny is based on the first TR dataset obtained by searching the UniProt database with XerD as a query and not the dataset with all of the homologues, which following parts of the paper refer to. This phylogeny is by definition XerD-biased and may lack a number of homologues.

2. The XerD-based phylogeny does not correspond to the results presented in the following sections which leads to some confusion. Some clarification on the following point might be helpful. The authors state that the presence of AB domain on the N-terminus of the TR is a good predictor of the mobile character of the TR as a part of ICE or phage integrase. However, the phage integrases were left out in ICE prediction. Could it be possible that the integrase is in 100kb from conjugative elements?

3. This study identifies 59 previously unannotated ICEs, which is quite an interesting aspect. Unfortunately, the authors did not include any experimental confirmation of the results nor any computational validation of the newly annotated ICEs.

Minor points:

1. There could be some improvement in the text to help readers follow it easier. The results and conclusions lack the straight, linear narrative and force the reader to jump between the sections of the paper. The Methods section seems too broad and could be partially moved to the results section to help the reader. For example, the result of the ICE prediction method bench-marking is not stated in Results section but in Methods.

2. Results section lacks the information on which part of the alignment was used to generate the phylogeny. This information should be mentioned before introducing the resulting tree. It is quite interesting as it indicates the strong phylogenetic signal was present in the core domains and the division is not simply due to presence or lack of the AB domain.

3. Figure 2 could be improved by drawing the phylogenetic relationships between the groups. The rectangles and arrows are shifted a bit below of the line on the graph for "Lambda".

4. About the node supports in phylogenetic trees: The Supplementary Figure 8 tree uses a different support value than the other trees. I understand that this may have been just an oversight, but for the careful reader, it may seem like presenting the best looking support. Moreover, none of the trees is presenting the traditional bootstrap support nor the nonparametric approximation. The fast approximation methods used in this study are known to generate high false-positive rate under severe model violations and should be generally supplemented with other methods (Anisimova M. et al. Syst Biol. 2011).

5. Supplementary Figure 7 could be made bigger, as in Supplementary Figure 6, this way the text can be more visible.

Reviewer #3:

This is a very interesting work, which might impact the Y recombinase field as much as the Nunes-Duby analysis of 105 Y rec published in 1998, did. It is interesting for both evolutionary biologists and people working on site specific recombination mechanisms. The phylogenic relationship with

Xer recombinases and the possible evolution history linking all the different Y recs which can be drawn from their data, reaches a level never attained so far, and will considerably help the classification, identification and proper annotation of YR and MGE in genomic data, as will be the AB logos they identified. The use of these data to identify and classify ICEs is very interesting as well, and their results (59 previously unannotated ICEs!) demonstrate how usefull it will be for improving proper annotation and discovery of new MGEs.

I only have minor comments :

- I suggest to use YR instead of TR, as TR was already given for terminal repeats in MGE, it will be less confusing.
- in figure 4B could you give the proportions of the different architecture schemes within each ICE type/ family, in a 4th column

in link with this, I am surprised that the author did not look for potential xis genes (directionality factors) sitting next to the Yrec genes in these ICEs, but also more broadly near the new YR that they identified and classified. They did not either discuss the existence of these directionality factors even if they are commonly used by many of the AB domain containing YR?

We sincerely thank the Reviewers for their constructive feedback. We are glad that they find our study interesting; their comments and advice have been very helpful in allowing us to improve the paper. We have now revised our manuscript according to the recommendations as specified below. Along with the recommended changes, we have also updated the manuscript to address all editorial queries and formatting issues.

Reviewer #1:

In this manuscript, the authors investigate classification of tyrosine recombinases in bacterial genomes. They provide a thorough analysis which is important, timely and interesting even for those not deeply involved in bacterial genomics. It's also written in a clear and accessible way, suitable for a general audience who may not be familiar with all of the methodological issues. I do have a few comments on the manuscript which I hope will improve it if attended to.

Major comments

The classification of the recombinases into subgroups, as illustrated in Figure 1A, looks very robust at least at the level of the major groups. Nevertheless, It is a slight concern here that the authors have picked one method and base all their conclusions on this. Alternatives should be tried to validate the robustness of the classification. Specifically

1) Can the authors investigate using simple blast as an alternative to jackhammer?

We chose to use jackhammer to aid the identification and inclusion of distantly related to XerD homologues in our dataset. It was previously reported that for proteins with less than 30% sequence identity, which is typical for our dataset, non-iterative pairwise BLAST finds fewer true homologs. For example, only one-half of the relatives with 20–30% identity was detected in the work of Brenner et al., PNAS, 1997. In contrast, methods that rely on information from multiple sequences, such as iterative jackhammer, do much better (see e.g. Park et al., JMB 1998). We now explain the advantages of jackhammer for the specific purpose of our study and refer to previously conducted comparative study in the revised manuscript (Page 17 Lines 17-19). Additionally, we included a description of the jackhammer approach in the text for further clarification, as suggested below.

2) The e-value cutoff is not explained. It seems to strike a balance between sensitivity and specificity but I think it would be a sound plan if the authors were to show that their classification is robust and that their cutoff optimises sensitivity and specificity. It would thus be good if the authors tried i) a more restrictive cutoff ii) a much more lenient cutoff (say, 1e-5)

We thank the reviewer for this comment. Indeed, the cut-off in the original manuscript was selected to optimize sensitivity and specificity of our search. As a general criterion, we aimed to include as many bacterial tyrosine recombinases (referred to as YR in the revised manuscript, in line with the reviewers' suggestions) as possible, while avoiding eukaryotic YRs. These proteins are known to be rather distant from bacterial YR, and poorly conserved, thus contributing to increased difficulties in reconstructing phylogeny.

In the current revision, we have systematically tested e-values from 1e-5 to 1e-40 and assessed the performance of these thresholds in including various recombinases, such as ICE and phage YRs, bacterial YRs, eukaryotic YRs. The results of this examination are presented now in the Appendix Table S2. In brief, we found that the threshold of 1e-30 retains more than 80% of all bacterial recombinases and discards 96% of eukaryotic YRs. Notably, all ICE integrases fall within this threshold (Dataset EV3). A more stringent cutoff of 1e-35 reduces the dataset to 68% of the bacterial YRs, while only slightly reducing the number of included eukaryotic YRs. A less stringent cutoff of 1e-25 also performs very well (89% bacterial YRs retained, 92% of eukaryotic YRs removed), while 1e-20 already includes significant number of eukaryotic YRs (only 82% of eukaryotic recombinases are removed). Therefore, we conclude that values from 1e-25 to 1e-30 are good compromise for our study design. We now include this analysis in the revised manuscript (Page 17 Lines 15-20 and Page 18 Lines 1-7).

3) Similarly, the hmmscan cutoff was $1e-20$, which is not consistent with the $1e-30$ cutoff used for the jackhammer- no explanation was given. Some investigation of the robustness of the results to this parameter would also be helpful.

Hmmscan was applied to map conjugation machinery proteins using profile HMMs developed in Abby et al., 2013 and supplied as part of the CONJscan tool. Given the distinct characteristic of these proteins and YRs, different E-value cut-offs are anticipated. In our original analysis we used a rather stringent cut-off value of $1e-20$, which was used for conservative identification of some conjugation system members in the original CONJscan paper (Abby et al., 2013). In the revised manuscript, we have rerun our analysis using a less strict cut-off ($1e-3$), which is recommended as default in CONJscan for all conjugation HMM profiles (Abby et al., 2013). This modified threshold did not lead to identification of additional ICEs compared to our original study. We now clarify our e-value choice on Page 20 Lines 29-31 of the revised manuscript.

The identification of conjugation machinery in the methods is described as being done by a "custom perl script" which is available through GitHub. It's great that the script is available but it would be important for the authors to also explain what the script is doing and how it is working in the methods section- it was not obvious to me how these proteins are identified and whether this relies on previous classification or not.

In the revised manuscript, we have extended explanation of the method (Page 20 Lines 21-31 and Page 21 Lines 1-5). In brief, our pipeline retrieves the genomic position of the recombinase, expands this position 100 kilobases upstream and downstream, and identifies conjugation machinery proteins within this genomic region using profile HMMs from the CONJscan database. We have also updated the script and improved our README on GitHub to improve clarity.

Minor comments

For future submissions, it might be worth including line numbers for ease of reference for reviewers.

Line numbers are included in the revised version.

Page 3

"Some MGEs hijack host-encoded Xer proteins[8,9] while others encode specific TRs to promote their own integration... "

Which part of the sentence does the reference refer to? Please add references if required for the second part of the sentence

The missing references have been added.

"TRs contribute to the emergence of antibiotic resistance on several ways..."
Should read "in several ways"

Corrected.

P4

"Building on the largely expanded..."
Not very clear. Might as well just remove the first subclause and instead begin the sentence:
"Here we assembled an extended... "

Done.

"To comprehensively analyse"

Change to "To analyse" - can't be sure that you've covered everything using this method [Or "to boldly analyse" if you really want a split infinitive here ☹]

The sentence has been changed as suggested by the referee.

"jackhammer search"

If you are to mention jackhammer in the main text I think you should give a brief explanation of what this is, and also a reference to the hmmer version where jackhammer was first introduced, similarly to the way you describe HMM on p5.

We included a description of the method and the corresponding reference on Page 5 Lines 4-11.

P5

"Two major phylogenetic groups..."

The figure looks very convincing but it would be worth explicitly stating the branch support values for these two groups in the text. Similarly, what are the branch support values for the subgroups in the following descriptions?

We now state the thresholds for the observed branch support values in the Results section (Page 5 Lines 17-19). Furthermore, we added Appendix Table S1, where we list support values for all relevant branches, and we supply the full tree, with an option for interactive inspection online (<http://itol.embl.de/shared/gera>).

P12

"we divide all TR proteins into two groups..." should be "we divided.."

Corrected.

"been experimentally shown" should be "been shown"

Changed.

Reviewer #2:

Summary:

Smyshlyaev et al. begin their sequence analysis of Tyrosine Recombinases (TRs) by searching the proteome database using the XerD protein from *E. coli*. Then they generate a phylogenetic tree and utilize HMM to find all TR homologues present in UniProt proteomes collection. A preceding maximum likelihood phylogenetic tree analysis for representative sequences from diverse tyrosine recombinase families was carried out by Wang J. et al. in 2018 (published in NAR). However, a further investigation of Xer recombinases was required, since it is clear that they have been recruited from MGEs. Here in this manuscript, Smyshlyaev et al. pave the way to the understanding of TRs with a more comprehensive analysis. The authors then extract the 50 most abundant TR proteins and characterize them further. This analysis reveals that all TRs can be classified in two major phylogenetic groups based on the presence or absence of the highly variable N-terminal Arm-Binding (AB) domain. Following this, the authors provide a detailed depiction of several sub-groups within these two major groups. These sections are descriptive sequence analysis of each subgroup of TR homologues and as well include the speculations on their function. Nevertheless, the focus on functional and conserved residues among the TRs within each subgroup leads to the novel insights. For instance, the commonly used prototype; lambda phage integrase is apparently quite different from other TRs. Another novel finding is the high correlation of the presence of AB domain and the mobile character of the TR, as a part of Integrative and Conjugative Elements (ICEs) or as phage integrases. This observation was followed up with the identification of new, putative ICEs by the presence of AB-containing TR in the genomic vicinity of conjugative elements. The method was bench-marked on ICEberg

database and proved to predict all full-length ICEs found in the database as well as 59 previously unannotated ICEs.

General Remarks:

The paper certainly presents a significant body of work, the tools used are solid and generally used properly. The conclusions drawn from the results sound convincing but leave some questions unanswered or not addressed. Overall, this study provides a thorough cataloguing of TRs which is potentially important for MGE identification and their impact on adaptation and will potentially serve as a resource for other interested scientists in the future. The main novelty is linking the presence of Arm Binding domain with the function in ICE or phage mobility and using it as a predictor for ICE annotation. The work presents an interesting basis for further development of sequence-based predictors of TR function.

There is a recent review by Koonin et al. (published in *Nat. Rev. Genetics* 21, pages 119-131 (2020)) on MGEs and to my surprise Smyshlyaev et al. did not cite this paper. If there was a detailed and broader outlook in addition to the discussion of this study following the points which were raised in Koonin's review, the significance of this study could be pronounced sufficiently.

We thank the reviewer very much for this comment. Indeed, we believe our study provides a new exciting angle on the 'domestication versus hijacking' dilemma in MGE evolution, which we now discuss on Page 15 Lines 1-4.

Major points:

1. The most conspicuous point of this work lies in the phylogenetic analysis of TR dataset. The presented phylogeny is based on the first TR dataset obtained by searching the UniProt database with XerD as a query and not the dataset with all of the homologues, which following parts of the paper refer to. This phylogeny is by definition XerD-biased and may lack a number of homologues.

To create a complete tyrosine recombinase sequence dataset for our phylogenetic reconstruction, we performed an iterative jackhmmer search. This consisted of several cycles of sequence searches, each querying with a new profile HMM created with the results of the previous search cycle. All resulting sequences were then clustered, and representatives of all the clusters were used for phylogeny reconstruction. Thus, the phylogenetic reconstruction covers the same diversity of sequences as all other parts of the paper. While the initial step of the search pipeline used XerD as a query, the final dataset also includes distant homologues and does not appear to carry an XerD bias. The iterative jackhmmer search was reported to perform well in identifying distantly related sequences and in the revised manuscript we find that our analysis recovers more than 80% of all the bacterial tyrosine recombinases (YRs) annotated in Pfam (Page 17 Lines 15-20 and Page 18 Lines 1-7). For more details regarding the specificity and sensitivity of our jackhmmer search, please also refer to our response to Reviewer 1, major point 2.

2. The XerD-based phylogeny does not correspond to the results presented in the following sections which leads to some confusion. Some clarification on the following point might be helpful. The authors state that the presence of AB domain on the N-terminus of the TR is a good predictor of the mobile character of the TR as a part of ICE or phage integrase. However, the phage integrases were left out in ICE prediction. Could it be possible that the integrase is in 100kb from conjugative elements?

Our analysis does not effectively discriminate between ICE and phage integrases. In fact, we find that many MGE-related subgroups contain both phage and ICE integrases (see for example, Int_{Tn916} and Int_{SXT} subgroups in Appendix Figures S6 and S7). As we used the hmm profiles corresponding to these mixed YR subgroups for ICE prediction, phage integrases were also included in our analysis. We have carefully checked the manuscript to clarify this point and avoid

confusion. We hope that our revisions address the referee's concern.

3. This study identifies 59 previously unannotated ICEs, which is quite an interesting aspect. Unfortunately, the authors did not include any experimental confirmation of the results nor any computational validation of the newly annotated ICEs.

We agree with the reviewer that additional evidence for the mobility of these elements will be valuable. However, experimental verification requires specific native bacterial strains, which are not easily accessible. Reconstitution of ICE mobility is also generally challenging, mostly due to limitations in bacterial culturing conditions, specific host factor requirements, and the complex regulatory landscape of these elements. Accordingly, only a few ICEs have been adequately characterised to date and analysis of a substantial number of our predicted ICEs would be beyond the scope of our current manuscript.

Thus, we have now extended our paper with three levels of computational verification. First, we predicted the borders of identified ICEs. Typically, ICE ends are marked with terminal repeats, which are recognised by the integrase enzyme to execute ICE excision and integration. The integrase gene is usually located adjacent to one of these terminal repeats. Thus, the identification of terminal repeats and mapping of their location relative to the integrase gene will support the identity of our newly annotated ICEs. With this analysis, we have successfully predicted terminal repeats for 50 out of 59 ICEs, as reported in Dataset EV4.

Second, we tested if a complete conjugation system is encoded within the predicted borders of the candidate ICEs and shows sequence conservation in closely related elements. Active ICEs generally include conjugation machinery genes, which are required for their intercellular transfer. The essentiality of these modules implies a high level of conservation within ICE families. We successfully detected conjugation systems in all predicted ICEs and found that closely related ICEs bear conjugation systems of the same type (see Figure 5 and Appendix Figure S8).

Third, we attempted to identify naïve (ICE-free) genomic sites for confirmed ICEs in closely related genomes. Given the mobile nature of ICEs, functional elements can move to new genomic sites and may thus appear in different locations across different bacterial strains or species. Thus, successful identification of naïve sites would provide direct confirmation for ICE identity and movement. Our analysis identified naïve insertion sites for 18 of the 50 confirmed ICEs (Dataset EV4), providing strong evidence of recent mobilisation of these ICEs. One illustrative example with ICE-containing and naïve genomes is highlighted in Appendix Figure S9. For the remaining elements we did not find a closely related ICE-free genome in the available genome sequence database. This is not unexpected, since most of the new ICEs are detected in previously unannotated genomes from underrepresented taxa.

These additional analyses provide strong support for the identity of the detected ICEs and hence verify the utility of our pipeline for MGE annotation. The corresponding analyses and results are now outlined in the Results section of the manuscript (Page 12 Lines 23-31 and Page 13 Lines 1-6) and in Methods (Page 21 Lines 6-24). We thank the referee for their comment; we believe that the additional analyses have largely enriched our manuscript.

Minor points:

1. There could be some improvement in the text to help readers follow it easier. The results and conclusions lack the straight, linear narrative and force the reader to jump between the sections of the paper. The Methods section seems too broad and could be partially moved to the results section to help the reader. For example, the result of the ICE prediction method bench-marking is not stated in Results section but in Methods.

We have thoroughly revised our manuscript to improve clarity. We have added narrative at the beginning of each section to clarify the flow. Additionally, we extended the Results section with general descriptions of the methodological concepts of our study, as suggested by the reviewers.

2. Results section lacks the information on which part of the alignment was used to generate the phylogeny. This information should be mentioned before introducing the resulting tree. It is quite interesting as it indicates the strong phylogenetic signal was present in the core domains and the division is not simply due to presence or lack of the AB domain.

That is indeed an important point and we are thankful to the referee for this comment. We have used only CB and CAT domain sequences for the alignment; thus, the final phylogenetic reconstruction is independent of the presence of the AB domain. We have added this information in the Results section (Page 5 Lines 13-14).

3. Figure 2 could be improved by drawing the phylogenetic relationships between the groups. The rectangles and arrows are shifted a bit below of the line on the graph for "Lambda".

We have updated the figure as suggested.

4. About the node supports in phylogenetic trees: The Supplementary Figure 8 tree uses a different support value than the other trees. I understand that this may have been just an oversight, but for the careful reader, it may seem like presenting the best looking support.

The tree shown in Appendix Figure S8 (previously Supplementary Figure 8) was reconstructed independently from the main tree, using only the sequences of ICE integrases. We now clarify this in the Results (Page 13 Lines 8-11) and Methods (Page 19 Lines 5-12) sections, as well as in the Figure legend.

Moreover, none of the trees is presenting the traditional bootstrap support nor the nonparametric approximation. The fast approximation methods used in this study are known to generate high false-positive rate under severe model violations and should be generally supplemented with other methods (Anisimova M. et al. *Syst Biol.* 2011).

To address this comment, we now support our phylogenetic reconstruction with non-parametric SH-aLRT, as recommended in Anisimova M. et al. (*Syst Biol.* 2011). The resulting support values are shown in Appendix Table S1. Notably, our phylogenetic reconstruction is further substantiated by the observed conservation of functionally relevant sequence features (such as host taxonomy and presence of specific DNA-binding domains) within the individual subgroups.

5. Supplementary Figure 7 could be made bigger, as in Supplementary Figure 6, this way the text can be more visible.

Done.

Reviewer #3:

This is a very interesting work, which might impact the Y recombinase field as much as the Nunes-Duby analysis of 105 Y rec published in 1998, did. It is interesting for both evolutionary biologists and people working on site specific recombination mechanisms. The phylogenic relationship with Xer recombinases and the possible evolution history linking all the different Y recs which can be drawn from their data, reaches a level never attained so far, and will considerably help the classification, identification and proper annotation of YR and MGE in genomic data, as will be the AB logos they identified. The use of these data to identify and classify ICEs is very interesting as well, and their results (59 previously unannotated ICEs!) demonstrate how usefull it will be for improving proper annotation and discovery of new MGEs.

I only have minor comments :

- I suggest to use YR instead of TR, as TR was already given for terminal repeats in MGE, it will be less confusing.

We agree with the reviewer and updated the nomenclature in the revised manuscript.

- in figure 4B could you give the proportions of the different architecture schemes within each ICE type/ family, in a 4th column

We thank the reviewer for this suggestion. We have updated the figure accordingly.

in link with this, I am surprised that the author did not look for potential *xis* genes (directionality factors) sitting next to the *Yrec* genes in these ICEs, but also more broadly near the new YR that they identified and classified. They did not either discuss the existence of these directionality factors even if they are commonly used by many of the AB domain containing YR?

We agree with the reviewer regarding the relevance of *xis* genes for YR function, which we now discuss in our revised manuscript. Annotation of *xis* genes near YRs could also help to support ICE identification and characterisation. However, Xis proteins are very short and poorly conserved, which hampers their sequence-based identification. Nevertheless, in this revision we created a collection of Pfam profile HMMs that correspond to known Xis proteins and searched for the Xis proteins in 1kb proximity from the integrases in the studied ICEs. We observe that only ICEs with AB-containing integrases (but not simple YRs from *Int_{KX}* subclade) encode putative Xis. This analysis is described in Methods (Page 20 Line 31 and Page 21 Lines 1-2), presented and discussed in the Results (Page 13 Lines 20-28), and depicted in Figure 4 and Appendix Figure S8.

Thank you again for sending us your revised manuscript. We have now heard back from reviewers #1 and #2 who were asked to evaluate your study. As you will see below, both reviewers are satisfied with the performed revisions and they are supportive of publication. Reviewer #2 only lists a remaining concern regarding the quality of Appendix Figure S8, which we would ask you to address in a minor revision.

Moreover, we would ask you to address some remaining editorial issues listed below.

REFEREE REPORTS

Reviewer #1:

I thank the authors for their careful and considerate response to my queries in the first round of review. I'm satisfied with their answers. It also appears that they have addressed the concerns of the remaining reviewers in a thoughtful manner and I would therefore support publication of the revised version.

Reviewer #2:

In the revised manuscript, Smyshlyaev et al. addressed and clarified the raised major points. More specifically:

The addition of jackhammer bench-marking convincingly shows that the data set used in the study

captures distant XerD homologues. I would like to thank the authors for clarifying my comment regarding the distinction between phage and ICE integrases.

One of the main ambiguities that was highlighted in my review was the lack of additional validation of the character of newly predicted elements. Smyshlyaev et al. now in the revised manuscript included a large body of computational analysis providing 3 independent points of evidence strongly suggesting that the predicted ICEs are indeed functional mobile elements. It is especially convincing in the case of 18 out of 59 putative ICEs for which the evidence of recent mobility was provided.

As for the minor points: The revised manuscript is written in a clear manner with a straightforward narrative. Most of the figures are legible with the exception of the appendix figure S8, of which the resolution is too low to read the labels and should be improved.

All the other minor points were addressed.

I would recommend this manuscript to be published in MSB after improving the image quality for the above mentioned figure.

Response to the Reviewers

Reviewer comments:

Reviewer #1:

I thank the authors for their careful and considerate response to my queries in the first round of review. I'm satisfied with their answers. It also appears that they have addressed the concerns of the remaining reviewers in a thoughtful manner and I would therefore support publication of the revised version.

Reviewer #2:

In the revised manuscript, Smyshlyaev et al. addressed and clarified the raised major points. More specifically:

The addition of jackhammer bench-marking convincingly shows that the data set used in the study captures distant XerD homologues. I would like to thank the authors for clarifying my comment regarding the distinction between phage and ICE integrases.

One of the main ambiguities that was highlighted in my review was the lack of additional validation of the character of newly predicted elements. Smyshlyaev et al. now in the revised manuscript included a large body of computational analysis providing 3 independent points of evidence strongly suggesting that the predicted ICEs are indeed functional mobile elements. It is especially convincing in the case of 18 out of 59 putative ICEs for which the evidence of recent mobility was provided.

As for the minor points: The revised manuscript is written in a clear manner with a straightforward narrative. Most of the figures are legible with the exception of the appendix figure S8, of which the resolution is too low to read the labels and should be improved.

All the other minor points were addressed.

I would recommend this manuscript to be published in MSB after improving the image quality for the above mentioned figure.

Response:

We thank both reviewers for their work on our manuscript and for their feedback. We are glad to hear that they find that our revised manuscript appropriately addressed their comments and support the publication of the work in *Molecular Systems Biology*.

We apologise for the poor quality of the Appendix Figure S8. We have replaced this figure with an improved quality version in the current submission.

Thank you again for sending us your revised manuscript. We are now satisfied with the modifications made and I am pleased to inform you that your paper has been accepted for publication.

Corresponding Author Name: Georgy Smyshlyaev

Journal Submitted to: MOLECULAR SYSTEMS BIOLOGY

Manuscript Number: MSB-20-9880R